# The Influence of the Q-Angle and Muscle Strength on Idiopathic Anterior Knee Pain in Adolescents

**DOI:** 10.3390/medicina59061016

**Published:** 2023-05-24

**Authors:** Darko Milovanović, Ninoslav Begović, Bojan Bukva, Siniša Dučić, Aleksandar Vlahović, Zoran Paunović, Marko Kadija, Nikola Topalović, Lazar Stijak

**Affiliations:** 1Clinic for Orthopedic Surgery and Traumatology, Clinical Center of Serbia, Pasterova 2, 11000 Belgrade, Serbia; darkomil@doctor.com (D.M.); kadija.marko@gmail.com (M.K.); 2School of Medicine, University of Belgrade, dr Subotića 8, 11000 Belgrade, Serbia; ninobego@hotmail.com (N.B.); bojanbukva@yahoo.com (B.B.); sinisaducic@gmail.com (S.D.);; 3Institute for Mother and Child Health Care of Serbia, Radoja Dakica 6-8, 11070 Belgrade, Serbia; 4University Children’s Hospital, Tiršova 10, 11000 Belgrade, Serbia; 5Institute of Medical Physiology, Faculty of Medicine, University of Belgrade, Visegradska 26/II, 11000 Belgrade, Serbia; 6Department for Anatomy, School of Medicine, University of Belgrade, dr Subotića 8, 11000 Belgrade, Serbia

**Keywords:** adolescent, knee pain, Q-angle, muscle strength

## Abstract

*Background and Objectives*: Idiopathic anterior knee pain is a common condition in adolescents and is mostly of unknown cause. The aim of this study was to examine the influence of the Q-angle and muscle strength on idiopathic anterior knee pain. *Materials and Methods*: Seventy-one adolescents (41 females and 30 males) diagnosed with anterior knee pain were included in this prospective study. The extensor strength in the knee joint and the Q-angle were monitored. The healthy extremity was used as a control. The Student’s paired sample *t*-test was applied for testing the difference. Statistical significance was set at 0.05. *Results*: There was no statistically significant difference in the Q-angle value between the idiopathic AKP and the healthy extremity (*p* > 0.05) within the entire sample. A statistically significant higher Q-angle of the idiopathic AKP knee (*p* < 0.05) was obtained in the female subgroup. No statistically significant difference (*p* > 0.05) was found in the male subgroup. Within the male subgroup, the strength of the extensors within the knee joint of the healthy extremity had statistically significant higher values than the strength of these muscles in the affected extremity (*p* < 0.05). *Conclusion*: A greater Q-angle is a risk factor linked to anterior knee pain within the female population. Decreased muscle strength of knee joint extensors is a risk factor linked to anterior knee pain in both sex subgroups.

## 1. Introduction

The term “anterior knee pain” (AKP) is suggested to encompass all pain-related problems of the anterior portion of the knee in the population of athletes and non-athletes. It is a common symptom, especially in adolescence [1,2]. In current literature, there is no consensus about the definition, classification, assessment, diagnosis, and management of AKP. Physicians who treat patients with AKP should understand the normal anatomic features and the biomechanics of the knee joint [1,2,3,4,5]. Several theories have been proposed, including muscular imbalance, excessive physical activity, and extremity malalignment. The etiology is multifactorial and comprises a complex relationship between the many anatomical, biomechanical, psychological, social, and behavioral variables. The manifestation of pain is the result of varied predisposing factors and pathoanatomy [1,6]. Chronic idiopathic AKP affects between 14% and 17% of the young active population and is typically exacerbated by activities [4]. For many years, the term chondromalacia patellae was accepted to denote a set of symptoms related to knee pain [3]. However, in a majority of adolescents with this condition, no specific cause can be found [1,7].

Different etiologic mechanisms of idiopathic AKP have been suggested, including patellar maltracking, chondromalacia patellae, quadriceps muscle imbalance, lower extremity malalignment, and increased physical activity [3,5]. There is no single factor causing pain, as it is believed to be the result of numerous pathophysiological processes [8]. The extrinsic causes involved in AKP are mainly mechanical and include wasting of the medial quadriceps and an increased quadriceps angle (Q-angle) [9].

The Q-angle is one of the most frequently studied static parameters in AKP but its clinical usefulness remains controversial [10]. The Q-angle was originally defined by Brattstrom. He described the Q-angle as an angle with its apex at the patella and formed between the ligament patellae and the extension of the line formed by the quadriceps femoris muscle resultant force, using the anterior superior iliac spine (ASIS) as the proximal landmark [11]. For men, the average Q-angle is 14°, and for women it is 17° [12]. Muscle strength is one of the most frequent dynamic indicators of AKP. Decreased knee extensor strength is a common finding in patients with AKP. Muscle tightness and muscular imbalance of the lower extremity muscles with decreased strength due to hypotrophy or inhibition have been suggested, but remain unclear, as potential causes of AKP [2].

## 2. Patients and Methods

A total of 71 patients with idiopathic AKP were evaluated over a period of 24 months at the Pediatric Orthopedic Surgery Clinic of the Institute for Mother and Child Health Care, in Belgrade, Serbia. The study included 41 girls (57; 75%) and 30 boys (42; 25%), with the average age of 15.4 ± 1.5 years (range: 12–18 years). Only patients with unilateral pain were observed. The pain involved the right knee in 39 patients (54.93%), and the left knee in 32 patients (45.07%). All participants had a history of knee pain lasting more than 6 months and had undergone radiography in the antero-posterior and lateral view, and magnetic resonance imaging (MRI) of the idiopathic AKP knee. The control group included the contralateral non-affected knee. Limb dominance is excluded as a risk factor. Patients who had undergone physical therapy prior to AKP, patients with a history of injury or surgery in the lower limb, and patients who presented with neurological, cardiovascular, or rheumatic disease were excluded from the study. In addition, patients with congenital anomalies of the lower limb were also excluded.

There are some limitations in the present study such as the control group, which is the contralateral non-affected knee. We believe that such a control group has less chance of error than a healthy population.

A questionnaire (survey) on demographic data was used to collect socio-epidemiological data and risk factor data. Survey questions were related to the level, type, and intensity of physical activities that patients were involved in. All patients (parents or guardians) signed the consent form for participation in this study.

The following parameters were evaluated: The Q-angle and the peak and the mean isometric strength of knee extensors (the quadriceps femoris muscle) in the idiopathic AKP and the non-affected knee (control group). The quadriceps angle (Q-angle) represents the angle between two lines: the first—from the anterior-superior iliac spine to the mid-patella, and the second—from the mid-patella to the tibial tubercle, and in the present study, it was measured with a standard goniometer. The measurement was taken in the supine position with the knees extended and a relaxed quadriceps, three times by three different examiners. All examiners had a high level of experience. The mean value of the angle was taken. A hand-held dynamometer (Chatillon MSC-500, AMETEK, Inc., Largo, FL, USA) was used to collect the peak and mean isometric strength values for lower extremity muscles, in kilograms. The measurement was taken in the prone position with the knees bent at 90 degrees. The value that the patient could retain for at least 3 s was recorded.

SPSS 11.0 (SPSS Inc., 233 South Wacker Drive, 11th floor, Chicago, IL 60606-6412, USA) was used for statistical analysis. The Kolmogorov–Smirnov test was used to test the normality of distribution. The Student’s *t*-test was used to test the differences. The χ^2^ test was used for non-parametric observation features (contingency tables).

The Ethical Committee of the Pediatric Surgery Clinic of the Institute for Mother and Child Health Care, in Belgrade, Serbia, approved the study.

## 3. Results

Based on the data collected, it was found that 61/71 (86%) of the respondents were actively engaged in some type of sports. Only 10/71 participants (14%) with idiopathic AKP were not involved in sports, thus belonging to the population of physically inactive adolescents. According to the types of sports, the participants predominantly engaged in basketball, volleyball, and folk dancing. As far as the training surface was concerned, hard surfaces were predominant, the most frequently used being hardwood floor, for 50/71 participants (70%).

According to pain analysis, the right knee was affected in 39/71 participants (54.93%) and the left knee in 32/71 (45.07%). There was no statistically significant difference in the distribution of the subjects as to the side affected (χ^2^ = 0.690, *p* = 0.406; *p* > 0.05).

The mean value of the Q-angle measured on the non-affected knee was 12.6 ± 4.2 degrees, while the Q-angle measured on the idiopathic AKP knee was 12.8 ± 4.2 (*p* = 0.733; *p* > 0.05). By testing the Q-angle values in the female subgroup, a statistically significant higher Q-angle of the idiopathic AKP knee was obtained, as compared to the non-affected knee (*p* = 0.041). In the male subgroup, no statistically significant difference was found between the Q-angle in the idiopathic AKP and the non-affected knee (*p* = 0.254). The results are presented in Figure 1.

The mean muscle strength of knee extensors in the non-affected knee was 21.0 ± 3.4, while the muscle strength of the idiopathic AKP knee was 17.6 ± 4.5. A statistically significant difference (*p* < 0.001) was obtained between the extensor muscle of the non-affected and the extensor muscle of the idiopathic AKP knee. When the patients were divided into male and female subgroups, a statistically significant difference was found between extensor muscle strength in the idiopathic AKP and the non-affected knee (male subgroup *p* < 0.001; female subgroup *p* < 0.001). The results are presented in Table 1.

## 4. Discussion

Idiopathic AKP is a common condition amongst adolescents. Despite various theories, this disorder is still not completely understood [13].

Every day, doctors encounter patients in whom the diagnosis of AKP is reached by the method of exclusion. The idea behind this research was related to the notion of finding the precise answer to the question as to what a child is suffering from. The question remains as to whether it is necessary to treat this type of pain as AKP or not. Nimon et al. concluded that in girls with AKP, clinical examination and standard radiographs were appropriate methods for excluding serious pathology. They predicted that 73% of such patients would eventually improve without treatment and that the 27% who did not improve would still have no evidence of significant structural disease of the knee, 16 years later [7].

Some of the patients from the present study were lost to follow-up during the course of the study. There are few studies with a long period of follow-up for patients with AKP who have undergone treatment. In two large studies where patients were trained to carry out physical therapy at home, complete recovery was achieved in 75–85% of the patients [14,15].

Out of 71 respondents included in the survey within the present study, the data received showed that most of them (61 patients) were active athletes. The majority (48 patients) trained 4 or more times per week, which is considered competitive training. Natari et al. tested 49 patients with AKP, most of whom (32 patients) were engaged in sports, while 17 were involved in competitive sports, and they trained at least 4 times per week [16]. Within the group of subjects from the present study, no statistically significant difference was found in the distribution of patients by sex and age. In other major studies conducted on a similar sample, gender distribution was similar [14]. Karlsson et al. had 34 girls out of 48 patients in their study, which was a statistically significant predominance of the female gender [15]. Sandown and Goodfellow studied a group of 54 adolescent girls with AKP, contrary to others who explored the males in the military population [17,18]. Some authors claim that the most obvious reason why women have more anterior knee pain than men is the difference in lower extremity orientation and alignment [19].

Females have a higher prevalence of increased femoral anteversion associated with an increased Q-angle. Q-angle measurement is often used to evaluate patellofemoral malalignment. Various sources report an increase in the Q-angle as a significant or minimal risk factor for the development of idiopathic AKP [20,21,22,23]. Although measurement of the Q-angle is the first step in the examination of knee symptomatology, there is no consensus in literature on the impact of this angle on AKP. Some authors have determined, by examination, that the Q-angle has no effect on AKP [21,22], while others claim that there is a correlation [23]. Hand and Spalding reported that the Q-angle measurement was a poor predictor of patellofemoral pain syndrome [17]. Q-angle measurement can be taken in different ways using different methods, such as the radiographic method, or clinically through the goniometer, and in different ways, having the patient in the supine position with the knee in total extension and relaxed quadriceps, or contracted, knee 30° of flexion with quadriceps relaxed, sitting with the knee in a 90° flexion and in a standing position [24]. In the present study, Q-angle measurement was performed in supination, with extended legs and a relaxed quadriceps. Freedman at al. conducted a study to determine how accurately the Q-angle represented the line-of-action of the quadriceps and whether adding active quadriceps contraction or the bent knee position to the measurement of the Q-angle would improve its reliability, accuracy, and association with patellofemoral kinematics [4]. They found that the Q-angle did not represent the line-of-action of the quadriceps. The average difference between each clinical and the MR-based Q-angle ranged from 5° to 8°. Adding an active quadriceps contraction or a bent knee position did not improve the reliability of the Q-angle. An increased Q-angle correlated to medial patellar displacement and tilt (r = 0.38–0.54, *p* < 0.001) in the cohort with anterior knee pain. They concluded that clinicians should not use the Q-angle to infer patellofemoral kinematics. The same method of measurement applied in the present study was applied by Silva et al. Static measurement was performed in 25 female patients with unilateral AKP. Measurement was carried out in supination, with completely relaxed quadriceps, just like in the present study. The difference between the two studies was in the degree of flexion (Silva et al. measured at a 15° angle of flexion) [25]. Q-angle static measurements were not different between the groups and proved to be values of low discriminatory capability. In the present study, when patients were divided into two groups, 30 males and 41 females, there was a statistically significant difference in the Q-angle values obtained by static measurement in the female subgroup (*p* = 0.041; *p* < 0.05). When testing the Q-angle in the male subgroup, a statistically significant difference (*p* = 0.254; *p* > 0.05) was not found. Based on these results, it can be concluded that the increase in the Q-angle in the female subgroup was a risk factor for the emergence of AKP.

The Q-angle measurement was carried out in the study on 80 samples of Indonesia students by Phatama et al. [26] They measured Q-angle on 40 females aged 18–25 with right-side AKP. The control group consisted of 40 females of the same age without pain. They determined by measuring that average right-leg Q-angle of the sample with AKP is 20.60 ± 1.26, while the right-leg Q-angle of the sample without AKP is 14.85 ± 0.99. Furthermore, the *p*-value from the Mann–Whitney test was <0.0001, which indicates a significant difference in Q-angle between samples with AKP and samples without AKP.

Kumar et al. [24] concluded that the patient with AKP has a higher Q-angle as compared to the normal population. So, individuals with a lower Q-angle are asymptomatic individuals. This conclusion was made by a comparison of 100 patients with 122 AKP aged 15–35 years and 100 without AKP. The authors found significant correlation between height and weight and Q-angle, also.

The other authors did not establish a statistically significant difference in the increase of the Q-angle and the occurrence of AKP [15,27,28].

In the present study, muscle strength in patients was tested using a digital dynamometer and tape. Muscle strength of the extended leg was measured and compared with the values for the healthy contralateral limb. Measurements were made in pronation with the knees flexed at 90 degrees. The value in kilos that the patients could hold for 3 s was taken as an average. In patients with AKP, weakness is often encountered [29]. The disadvantage of the present study is the measurement of muscle strength after the onset of pain. More objective results would have been obtained by measuring strength before the onset of pain.

Muscular imbalances accompanied by decreased torque due to atrophy or inhibition of the lower extremity muscles have been suggested as a potential cause of AKP [30]. Similarly, to the present study, Kaya et al. determined isokinetic torque of the quadriceps muscle bilaterally at an angular velocity of 60° and 180°/s, by using an Isomed 2000 (D&R GmbH, Gewerbering, Hemau, Germany) isokinetic dynamometer. The test was performed with the patients seated at 70° hip flexion (from the supine position) and the knee angle at 90° flexion. They performed isokinetic measurement, as opposed to the dynamic measurement of muscle strength in the present study. There was a significant difference in the quadriceps peak torque at 60°/s (*p* = 0.01), (percentage difference 29%) with no significant difference at 180°/s between the affected and the unaffected side (percentage difference 8.5%) in the Kaya et al. study. Similarly, to this study, there was also a statistically significant difference between the strength of the extensor muscle of the healthy and of the affected leg in both groups, male and female (*p* < 0.001), in the present study. Boling et al. tried to determine the biomechanical risk factors for AKP. The specific factors examined were lower extremity kinematics and kinetics during a jump landing task, the Q-angle, the navicular drop, and the strength of the hip and knee muscles (hip abductors, hip extensors, hip external rotators, hip internal rotators, knee flexors, and knee extensors). They confirmed the hypothesis of an association between the development of PFPS and altered kinematics and kinetics, an increased Q-angle, an increased navicular drops, and decreased lower extremity strength [31]. The study was limited due to the specific grouping which the research was based on (a military population) and the age that was identical in all of the subjects. In relation to the selection of respondents, the present study represents a heterogeneous group, with respect to age and gender. The results obtained in the present study have also shown that decreased lower extremity strength is a statistically significant risk factor for the emergence of AKP. In the study performed by Callaghan and Oldham, there was no statistically significant difference in the muscle strength between the two legs (57 patients, 35 females and 22 males) [32]. Giles et al. tested muscular volume using CT and NMR imaging. They proved the presence of muscle atrophy in patients with AKP and the importance of strengthening exercises for loss of pain [33].

Correlation between the Q-angle and the knee strength was discussed by Sac et al. [34] in their study on 50 healthy and right-leg dominant men, mean age 22.3 ± 2.3 years; range, 18 to 27 years with a Q-angle between 5° and 20° (mean Q-angle: 13.0 ± 6.4 degree). They tested isokinetic muscle strength using a Humac Norm dynamometer. It has been established that an increased Q-angle causes the knee joint torque and power values to decrease, showing the angle to have a negative correlation with the resulting peak torque; thus, it has an important effect on the development of the peak torque. The investigations were only performed on healthy athletes; it is not clear whether a sedentary or patient with AKP would show the same amount of isokinetic strength and muscle activity during these measurements.

## 5. Conclusions

An increase in the Q-angle and muscular weakness of the extensors of the knee are risk factors in females with idiopathic AKP. Muscular weakness of the extensors of the knee is a risk factor in males with idiopathic AKP.

## Figures and Tables

**Figure 1 medicina-59-01016-f001:**
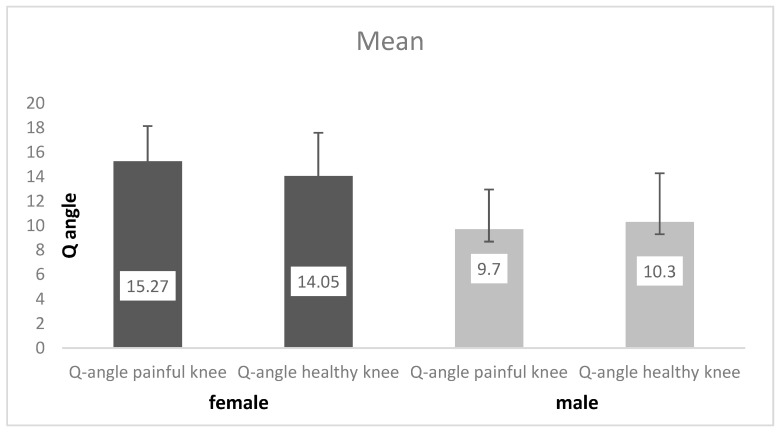
Q-angle values related to gender and status of the knee.

**Table 1 medicina-59-01016-t001:** Comparison of the muscular strength between the idiopathic AKP and the non-affected knee, by gender.

	Muscle Strength	Mean	Number	Standard Deviation
Male subgroup	Idiopathic AKP	18.983	30	3.4876
	Non-affected knee	22.517	30	2.2456
Female subgroup	Idiopathic AKP	16.634	41	4.9548
	Non-affected knee	19.902	41	3.6318

## Data Availability

All data of this study can be obtained by request to the following email: ninobego@hotmail.com.

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
