# Peer review of "The Influence of the Q-Angle and Muscle Strength on Idiopathic Anterior Knee Pain in Adolescents"

_medicina, 2023, doi:10.3390/medicina59061016_

Round 1

Reviewer 1 Report

Dear authors,

I am pleased to review the submitted paper medicina-2356302 entitled "The influence of the Q-angle and muscle strength on idiopathic anterior knee pain in adolescents"

The present paper focuses on influence of the Q-angle and muscle strength on idiopathic anterior knee pain in adolescents with different age, gender and pain side.

In my opinion the content is original, current, but not objective and persuasive.

1. Methods:” A total of 71 patients with idiopathic AKP were evaluated over a period of 24 months 58 at the Pediatric Orthopedic Surgery Clinic of the Institute for Mother and Child Health 59 Care, in Belgrade, Serbia.” Inclusion criteria of this study is unclear, and the diagnosis standard of idiopathic AKP is absence. The Exclusion criteria is also missing.

2. Methods: “Measurement was done in the prone position 82 with the knees bent at 90 degrees.” Q angle is the core index of this study, the measurement should be very critical. How many times did the authors measure the Q angle? What is the experience level of the examiners? Why the author did not measure the Q angle in supine or upright position.

3. Methods: “The control group included the 65 contralateral, non-affected knee.” Since the study choosed the affected knee as subject, the control group setting need to be explain, there is difference between dominant side lower limb with un-dominant side, the non- affected knee maybe exist anatomic abnormal but without symptoms because it is un-dominant side. This should be specified.

4. Results: “Figure 1. Q-angle values related to gender and status of the knee.” What is the difference between the first Q-angle painful knee and the second Q-angle painful knee, there is no  figure legend to give a detailed explaination.

5. Conclusions: “An increase in the Q-angle and muscular weakness of the extensors of the knee are 222 risk factors in females with AKP. Muscular weakness of the extensors of the knee is a risk 223 factor in males with AKP.” The study was focus on idiopathic AKP and the patient included is diagnosed as idiopathic AKP, it is not preciseness confused AKP with idiopathic AKP.

 Minor editing of English language required

Author Response

  1. The inclusion criteria and diagnostic standard of idiopathic AKP of this study are added in the lines 78-83.  The exclusion criteria of this study are added in the lines 84-88.  
  2. The measurement of the Q angle is additionally described in the lines 98-99.
  3. Limb dominance is excluded as a risk factor. (Line 84) If we had taken into account the dominance of the limb, as a control we would have had to take other persons, the morphological characteristics of the lower limb which are more different than the contralateral limb which is not dominant.

  4.  An additional explanation is placed in figure legend.
  5. The conclusion is aligned with the terms defined at the beginning.

Reviewer 2 Report

The framework of manuscript is intact and English writing is fluent. Although this issue is important and interesting, the definite patho-mechanisms to cause IAKP still cannot be clarified convincingly. All techniques used for study are challenged persistently in the literature. Therefore, the methodology used in this study is equivocal. Many references cited in this manuscript are out of date and many concepts with techniques have been quite different.

        Some doubts require clarification:

1.     In lines-24,27: In IAKP adolescents, is knee extensor or flexor muscle strength decreased? Table 1 reports extensor muscles. The conclusion in Abstract and Discussion section is contradicted.

2.     In line-48: The Q-angle should be fully spelled out when it appears first in the text.

3.     In line-52: The first mention of the Q-angle in the literature is Brattström (Acta Orthop Scand 1964), not Hungerford.

4.     In line-77: This step is the most critical situation for Q-angle measurement. It must be described (like line-159). The optimal technique to measure the exact Q-angle is continuously doubted: supine or standing, relaxed or contracted quadriceps femoris, or straight or flexed knee? Knee positions can significantly affect the measured degrees of the Q-angle. Consequently, the comparison of both knees become unbelievable. The statistical significance is therefore doubted. esp., a very small sample size: 30 males and 41 females.

5.     In lines-101,103, -- etc.: P-value must be exactly reported, not p > 0.05 simply.

6.     In line-108: Currently, the optimal technique to measure the Q-angle is still debated, and in this study the difference between both knees is only 1.22 degrees in females (Fig. 1). Do you believe that such a minimal difference can affect the initiation of IAKP?

7.     In line-122: AKP is initially used in line-32. It should be used thereafter in the whole text.

8.     In line-153: The cited reference is published in 1990 (ref. 20). Now, the concept is quite different. The normal range of the Q-angle has a minimal effect on influencing patellar malalignment. The references should be updated (refs. 21-23).

9.     In line-164 (ref.4): Freeman in 2014 doubted the representation between the Q-angle and line-of-action of quadriceps. It is due to failed measurement the Q-angle accurately. In 2020, a technique to accurate determination of the Q-angle was published (Orthop Surg 2020;12: 1270-6). Do you think whether the relationship for both will be different?  

10.  In line-211: The largest limitation in this study should be a small sample size (71 cases being divided to two groups) with an equivocal measured technique for the Q-angle. It can cause a completely contradicted conclusion.

11.  Why increased knee extensor strength may improve IAKP? In 2009 and 2018, two studies were published and the reported techniques may enforce muscle power of the vastus medialis obliquus (Arch Orthop Trauma Surg 2009; 129: 333-41; Gait & Posture 2018; 62: 440-4). 

Author Response

Dear reviewer, we have incorporated all your suggestions and corrected the text accordingly.

  1.  We apologize for the omissions in the abstract, which are the result of a translation error. 
  2. Corrected.
  3. We could not find the specified reference. Would you be so kind as to direct us to the link where we can find the said paper.
  4. Corrected (lines 98-99).
  5. Corrected (p=0.406; line 134).
  6. All patients were measured using the same technique by three researchers. We may not believe that that 1.22 is crucial for idiophatic AKP, but statistical tests indicate it.
  7. Corrected in whole text.
  8. Corrected (lines 181, 182).
  9. The idea of ​​a difference in the Q angle and line-of-action of quadriceps makes sense, but we did not have the ability to measure the line-of-action of quadriceps.
  10. I agree.
  11. The vastus medialis obliquus of the quadriceps is evolutionarily the youngest and the first to atrophy in an inactive knee. Consequently, by increasing the strength of the quadriceps, the strength of the vastus medialis obliquus which centralizes the patella, increases.

Round 2

Reviewer 1 Report

Dear authors,

I am pleased to review the revised paper medicina-2356302 entitled "The influence of the Q-angle and muscle strength on idiopathic anterior knee pain in adolescents"

The present paper focuses on influence of the Q-angle and muscle strength on idiopathic anterior knee pain in adolescents with different age, gender and pain side.

I reviewed this paper before and the authors have now submitted a revised version. I read the authors' response to the reviewer comments, the following questions I raised need to be appropriate answered. 

1. Methods:” A total of 71 patients with idiopathic AKP were evaluated over a period of 24 months 58 at the Pediatric Orthopedic Surgery Clinic of the Institute for Mother and Child Health 59 Care, in Belgrade, Serbia.” Inclusion criteria of this study is unclear, and the diagnosis standard of idiopathic AKP is absence. The Exclusion criteria is also missing.

Appropriate revised.

2. Methods: “Measurement was done in the prone position 82 with the knees bent at 90 degrees.” Q angle is the core index of this study, the measurement should be very critical. How many times did the authors measure the Q angle? What is the experience level of the examiners? Why the author did not measure the Q angle in supine or upright position.

Appropriate revised except "What is the experience level of the examiners?"

3. Methods: “The control group included the 65 contralateral, non-affected knee.” Since the study choosed the affected knee as subject, the control group setting need to be explain, there is difference between dominant side lower limb with un-dominant side, the non- affected knee maybe exist anatomic abnormal but without symptoms because it is un-dominant side. This should be specified.

There is no enough explaination why this study use contralateral limb as the control group. Actually the contralateral limb is not present symptons can not equal to a normal knee. In most cases the develoement of both limb are mirrored, the contralateral limb are non-affected maybe just because the contralateral limb are un-dominant side.

4. Results: “Figure 1. Q-angle values related to gender and status of the knee.” What is the difference between the first Q-angle painful knee and the second Q-angle painful knee, there is no  figure legend to give a detailed explaination.

After the revise the Figure 1 is still confusing, Q-angle should be present at Y axis not at X axis?

5. Conclusions: “An increase in the Q-angle and muscular weakness of the extensors of the knee are 222 risk factors in females with AKP. Muscular weakness of the extensors of the knee is a risk 223 factor in males with AKP.” The study was focus on idiopathic AKP and the patient included is diagnosed as idiopathic AKP, it is not preciseness confused AKP with idiopathic AKP.

Appropriate revised.

Minor editing of English language required.

Author Response

Dear reviewer, we have incorporated all your suggestions and corrected the text accordingly.

2. Corrected

3. We incerted explanation in methodology (lines 78-80).

4. Figure 1. is corrected (Q angle is on Y axis)

Reviewer 2 Report

After revision, this manuscript has been greatly improved.

However, some doubts still require re-checking:

1.     In line 53: Ref. 11 is cited. Brattström should the creator of the Q-angle [Shape of the intercondylar groove normally and in recurrent dislocation of patella. A clinical and X-ray-anatomical investigation. Acta Orthop Scand 1964; 68(Suppl): S1-148].

2.     In line 106: p= ?

3.     In line 126: Idiopathic AKP should be used.

4.     In Refs. 3,7,12,19: Lowercase should be used for the title of articles.

Author Response

Dear reviewer, we have incorporated all your suggestions and corrected the text accordingly.

  1. Reference is corrected.
  2. P value is incerted.
  3. Abbreviation is incerted.
  4. References are corrected.

Round 3

Reviewer 1 Report

Dear authors,

I am pleased to review the revised paper medicina-2356302 entitled "The influence of the Q-angle and muscle strength on idiopathic anterior knee pain in adolescents"

The present paper focuses on influence of the Q-angle and muscle strength on idiopathic anterior knee pain in adolescents with different age, gender and pain side.

I reviewed this paper before and the authors have now submitted a second revised version. I read the authors' response to the reviewer comments, the following questions I raised have been appropriate answered. 

My conclusion is accept.